# Precise predictions for boosted Higgs production

Kathrin Becker[1], Fabrizio Caola[2], Andrea Massironi[3], Bernhard Mistlberger[4],
Pier F. Monni[5], Xuan Chen[6,7], Stefano Frixione[8], Thomas Gehrmann[7], Nigel Glover[9],
Keith Hamilton[10], Alexander Huss[5], Stephen P. Jones[9], Alexander Karlberg[5],
Matthias Kerner[11], Kirill Kudashkin[12], Jonas M. Lindert[13], Gionata Luisoni[14],
Michelangelo L. Mangano[5], Stefano Pozzorini[7], Emanuele Re[15,16],
Gavin P. Salam[2,17], Eleni Vryonidou[18] and Christopher Wever[19]

## Abstract

Inclusive Higgs boson production at large transverse momentum is induced by different production channels. We focus on the leading production mechanism through gluon fusion, and perform a consistent combination of the state of the art calculations obtained in the infinite-top-mass effective theory at next-to-next-to-leading order (NNLO) and in the full Standard Model (SM) at next-to-leading order (NLO). We thus present approximate QCD predictions for this process at NNLO, and a study of the corresponding perturbative uncertainties. This calculation is then compared with those obtained with commonly used event generators, and we observe that the description of the considered kinematic regime provided by these tools is in good agreement with state of the art calculations. Finally, we present accurate predictions for other production channels such as vector boson fusion, and associated production with a gauge boson, and with a $t\bar{t}$ pair. We find that, at large transverse momentum, the contribution of other production modes is substantial, and therefore must be included for a precise theory prediction of this observable.

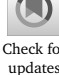
doi:10.21468/SciPostPhysCore.7.1.001

**1** Department of Physics, University of Warwick, Coventry, CV4 7AL, UK
**2** Rudolf Peierls Centre for Theoretical Physics,Oxford University, OX1 3PU, UK
**3** INFN, Sezione di Milano-Bicocca, Piazza della Scienza 3, 20126 Milano, Italy
**4** SLAC National Accelerator Laboratory, Stanford University, Stanford, CA 94039, USA
**5** CERN, Theoretical Physics Department, CH-1211 Geneva 23, Switzerland
**6** School of Physics, Shandong University, Jinan, Shandong 250100, China
**7** Physik-Institut, Universität Zürich, Winterthurerstrasse 190, CH-8057 Zürich, Switzerland
**8** INFN, Sezione di Genova, Via Dodecaneso 33, I-16146, Genoa, Italy
**9** Institute for Particle Physics Phenomenology, Department of Physics,
University of Durham, Durham, DH1 3LE, UK
**10** Department of Physics and Astronomy, University College London, London, WC1E 6BT, UK
**11** Institute for Theoretical Physics, Karlsruhe Institute of Technology,
76128 Karlsruhe, Germany
**12** Tif Lab, Dipartimento di Fisica, Università di Milano and INFN,
Sezione di Milano, Via Celoria 16, I-20133 Milano, Italy
**13** Department of Physics and Astronomy, University of Sussex, Brighton BN1 9QH, UK
**14** Max-Planck-Institut für Physik, Föhringer Ring 6, 80805 München, Germany

**15** Dipartimento di Fisica G. Occhialini, Università degli Studi di Milano-Bicocca and INFN,
Sezione di Milano-Bicocca, Italy
**16** LAPTh, Université Grenoble Alpes, Université Savoie Mont Blanc,
CNRS, 74940 Annecy, France
**17** All Souls College, Oxford OX1 4AL, UK
**18** Department of Physics and Astronomy, University of Manchester,
Oxford Road, Manchester M13 9PL, United Kingdom
**19** Physik-Department T31, Technische Universität München,
James-Franck-Strasse 1, D-85748 Garching, Germany

## Contents

## 1 Introduction

The continuously increasing amount of data recorded at the LHC opens the possibility to explore properties of the Higgs boson in a multitude of kinematic regimes. Of particular interest is the transverse momentum distribution of the Higgs boson for very large transverse momenta. Measurements of this observable allow for unique insights into the microscopic structure of the interactions of the Higgs boson with strongly interacting particles and might shed light on physics beyond the Standard Model. The observation of the Higgs boson in this kinematic regime is however extremely challenging.

The inclusive search for the Standard Model Higgs boson produced at large transverse momentum ($p_\perp$), and decaying to a bottom quark-antiquark pair, has been performed using data collected in pp collisions at $\sqrt{s} = 13$ TeV by the CMS and ATLAS experiments [1–6].

It is the objective of this document to study accurate theoretical predictions for the transverse momentum distribution with $p_\perp > 400$ GeV. We present new, state of the art predictions for the dominant gluon-fusion induced production of a Higgs boson and at least one hard partonic jet that recoils against it, based on perturbative QCD computations at a centre-of-mass energy of 13 TeV. In particular we perform a combination of next-to-next-to leading order (NNLO) calculations in the heavy top quark effective theory [7–13] with next-to-leading order (NLO) predictions in the full SM with finite top-quark mass ($m_t$) [14–16]. A related combination has been recently presented in ref. [17]. We provide a recommendation for the theoretical

prediction for the gluon-fusion channel to be used by the ATLAS and CMS collaborations. Subsequently, we compare these predictions with state of the art hard-event generators [18–22]. We find that indeed the most advanced event generators describe the cross sections of interest within uncertainties. Furthermore, we also report the contributions from the vector boson fusion, VH, and $t\bar{t}H$ production modes for the observable under consideration, together with a NLO calculation of the electro-weak corrections.

## 2 Predictions for the gluon-fusion channel

We start by focusing on the predictions for the gluon-fusion (ggF) channel, and by giving an approximate NNLO result, which we quote as our recommendation for the cross section in the boosted regime. This is obtained by combining the following two predictions for the production of a Higgs boson and at least one partonic jet: the NNLO ($\mathcal{O}(\alpha_s^5)$) calculation in the large-$m_t$ limit and the NLO ($\mathcal{O}(\alpha_s^4)$) calculation in the full SM.

The setup used for the NNLO results in the large-$m_t$ limit is as follows

- pp collisions at $\sqrt{s} = 13$ TeV,

- $m_H = 125$ GeV, $m_t = 173.2$ GeV, all other parameters as per YR4 [23],

- `PDF4LHC15_nnlo_mc`,

- central scales $\mu_F = \mu_R = M_{T,H}$, where we defined the Higgs transverse mass

$$M_{T,H} = \sqrt{m_H^2 + p_\perp^2}. \tag{1}$$

- In our predicions we consider an on-shell Higgs boson, so we do not include any particular decay.

In Section 2.2, we also consider the predictions from common event generators. Such predictions come with their own scale setting, as reported in the discussion below. The above scale choice is of course not unique, and different choices lead to differences in the final predictions. However, the goal of this manuscript is to compare different theory predictions for the observable under study. Therefore, we limit ourselves to the above choice for the discussion that follows.

### 2.1 Fixed-order

In this section we present state of the art predictions for the transverse momentum ($p_\perp$) spectrum of the Higgs boson in the boosted regime. The transverse momentum distribution was computed at NNLO in perturbative QCD in the heavy top quark effective theory (EFT) in refs. [7, 8, 10, 11]. Specifically, refs. [7, 8, 10, 11] compute NNLO corrections to the Born level production of a Higgs boson and a jet. In the EFT approximation the top quark is treated as infinitely heavy and its degrees of freedom are integrated out. It is however well known that the pure EFT computation fails to describe the $p_\perp$ spectrum for transverse momenta larger than $\sim 200$ GeV, cf. [15].

One way to improve on the pure EFT computation is to create the so-called Born-improved EFT approximation. To this end the EFT cross section is simply rescaled by the exact leading order SM cross section [24, 25]. For the inclusive (cumulative) cross section, defined as

$$\Sigma(p_\perp^{\text{cut}}) = \int_{p_\perp^{\text{cut}}}^{\infty} \frac{d\sigma}{dp_\perp'} dp_\perp', \tag{2}$$

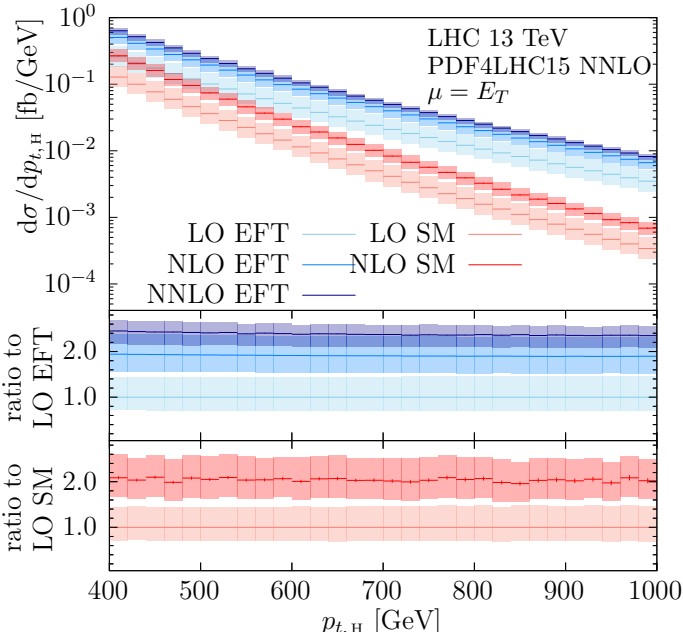

Figure 1: Transverse momentum distribution of the Higgs boson at the LHC with $\sqrt{s} = 13$ TeV computed in refs. [10, 15, 17]. The upper panel shows absolute predictions at LO ($\mathcal{O}(\alpha_s^3)$) and NLO ($\mathcal{O}(\alpha_s^4)$) in the full SM and in the infinite $m_t$ approximation (EFT), as well as the NNLO ($\mathcal{O}(\alpha_s^5)$) in the EFT. The lower panels show the ratio of the EFT and full SM predictions to their respective LO calculations. The bands indicate theoretical errors obtained with a 7-point scale variation, i.e. we perform a variation of $\mu_R$ and $\mu_F$ by a factor of two around their central value by keeping $1/2 \leq \mu_R/\mu_F \leq 2$.

this amounts to defining

$$\Sigma^{\text{EFT-improved (0), NNLO}}(p_\perp^{\text{cut}}) \equiv \frac{\Sigma^{\text{SM, LO}}(p_\perp^{\text{cut}})}{\Sigma^{\text{EFT, LO}}(p_\perp^{\text{cut}})} \Sigma^{\text{EFT, NNLO}}(p_\perp^{\text{cut}}). \qquad (3)$$

The numerical implications of this Born-improved NNLO predictions were first studied in ref. [10] and show deviations from the pure EFT computation at the level of 50% for transverse momenta of 400 GeV. Since this modification is performed at leading order, a considerable perturbative uncertainty has to be associated with this procedure and higher order corrections are desirable. In order to further improve the result several approximations were considered in refs. [21, 26–28] including exact real matrix elements at NLO in QCD and approximations for virtual matrix elements. Finally, the two-loop virtual matrix elements were included through an asymptotic expansion in refs. [14, 29], and exactly in refs. [15–17], hence allowing for the computation of the full NLO corrections. The exact NLO QCD corrections computed in refs. [15–17] modify the exact leading order prediction significantly but in a uniform way for the dynamical scale chosen here, as it can be appreciated from Fig. 1, from which one can observe a $K$ factor with a very mild $p_\perp$ dependence. An analogous behaviour is observed in the predictions obtained within the EFT. As a consequence, the modifications of the shape of the $p_\perp$ distribution of the Higgs boson due to finite $m_t$ effects is to a good extent already accounted for in Eq. (3) by the inclusion of exact leading order matrix elements. We collect in Table 1 the inclusive cross section $\Sigma$ for some relevant $p_\perp$ cuts up to both NNLO in the EFT [10] and to NLO in the full SM [15, 17]. We will adopt the predictions from these two references in the following study.

Ideally, we want to combine the NNLO predictions computed in the EFT with the exact NLO prediction. Under the assumption that the exact NNLO QCD corrections follow the pattern of the NNLO EFT corrections, i.e. they would lead to a uniform K-factor, this can be achieved by rescaling EFT NNLO predictions in the following way:

$$\Sigma^{\text{EFT-improved (1), NNLO}}(p_\perp^{\text{cut}}) \equiv \frac{\Sigma^{\text{SM, NLO}}(p_\perp^{\text{cut}})}{\Sigma^{\text{EFT, NLO}}(p_\perp^{\text{cut}})} \Sigma^{\text{EFT, NNLO}}(p_\perp^{\text{cut}}). \tag{4}$$

We quote the prediction obtained with Eq. (4) as the current best prediction.[1] To estimate the theory uncertainty in the resulting cross section we proceed as follows:

- We perform a variation of $\mu_R$ and $\mu_F$ by a factor of two around their central value by keeping $1/2 \le \mu_R/\mu_F \le 2$ (7 point scale variation). The scales are varied separately in $\Sigma^{\text{EFT, NNLO}}$ and in the $\Sigma^{\text{SM, NLO}}/\Sigma^{\text{EFT, NLO}}$ ratio. For the latter, the same scale is chosen for the numerator and the denominator, and the final uncertainty is symmetrised. Finally, the two uncertainties are combined either in quadrature or linearly.

- We assume that the uncertainty due to mass effects in the NNLO EFT correction is obtained by rescaling the latter by the relative mass correction at NLO. Thus, we assess the

---

[1]We point out that the rescaling performed in Eqs. (3), (4) could be alternatively defined at the differential level, leading to yet another prescription to combine consistently the NNLO prediction in the EFT with the NLO calculation in the full SM. Since in this document we will only refer to the cross section $\Sigma(p_\perp^{\text{cut}})$ we choose to perform the rescaling at the level of the cumulative cross section.

Table 1: Inclusive cross sections in fb and $K$-factors for $pp \to H + X$ in the SM for the relevant $p_\perp^{\text{cut}}$ values (in GeV units) as computed in refs. [10, 15, 17]. Uncertainties are estimated by varying $\mu_F$ and $\mu_R$ separately by factors of $1/2$ and $2$ while keeping $1/2 \le \mu_R/\mu_F \le 2$. The $K$-factors are defined as $K_{\text{SM}}^{\text{NLO}} = \text{NLO}_{\text{SM}}/\text{LO}_{\text{SM}}$, $K_{\text{EFT}}^{\text{NLO}} = \text{NLO}_{\text{EFT}}/\text{LO}_{\text{EFT}}$, and $K_{\text{EFT}}^{\text{NNLO}} = \text{NNLO}_{\text{EFT}}/\text{NLO}_{\text{EFT}}$.

Inclusive cross sections ([fb]) and $K$-factors for $pp \to H + X$

| $p_T^{\text{cut}}$ | $\text{LO}_{\text{full}}$ | $\text{NLO}_{\text{full}}$ | $K_{\text{full}}^{\text{NLO}}$ | $\text{LO}_{\text{EFT}}$ | $\text{NLO}_{\text{EFT}}$ | $\text{NNLO}_{\text{EFT}}$ | $K_{\text{EFT}}^{\text{NLO}}$ | $K_{\text{EFT}}^{\text{NNLO}}$ |
|---|---|---|---|---|---|---|---|---|
| 400 | $11.9^{+45\%}_{-29\%}$ | $24^{+24\%}_{-20\%}$ | 2.06 | $32^{+44\%}_{-29\%}$ | $63^{+23\%}_{-19\%}$ | $78^{+9.2\%}_{-12\%}$ | 1.93 | 1.25 |
| 450 | $6.5^{+45\%}_{-29\%}$ | $13.3^{+24\%}_{-20\%}$ | 2.05 | $21^{+45\%}_{-29\%}$ | $41^{+22\%}_{-19\%}$ | $51^{+8.9\%}_{-11\%}$ | 1.92 | 1.25 |
| 500 | $3.7^{+45\%}_{-29\%}$ | $7.5^{+24\%}_{-20\%}$ | 2.05 | $14.2^{+45\%}_{-29\%}$ | $27^{+22\%}_{-20\%}$ | $34^{+8.8\%}_{-11\%}$ | 1.91 | 1.25 |
| 550 | $2.1^{+45\%}_{-30\%}$ | $4.4^{+24\%}_{-20\%}$ | 2.04 | $9.8^{+45\%}_{-29\%}$ | $18.6^{+22\%}_{-20\%}$ | $23^{+8.8\%}_{-11\%}$ | 1.91 | 1.25 |
| 600 | $1.28^{+46\%}_{-30\%}$ | $2.6^{+24\%}_{-20\%}$ | 2.03 | $6.8^{+45\%}_{-29\%}$ | $13.0^{+22\%}_{-20\%}$ | $16.2^{+8.8\%}_{-11\%}$ | 1.90 | 1.24 |
| 650 | $0.79^{+46\%}_{-30\%}$ | $1.60^{+24\%}_{-20\%}$ | 2.03 | $4.9^{+46\%}_{-29\%}$ | $9.3^{+22\%}_{-20\%}$ | $11.5^{+8.7\%}_{-11\%}$ | 1.90 | 1.24 |
| 700 | $0.49^{+47\%}_{-30\%}$ | $1.00^{+24\%}_{-20\%}$ | 2.03 | $3.5^{+46\%}_{-29\%}$ | $6.7^{+22\%}_{-20\%}$ | $8.3^{+8.7\%}_{-11\%}$ | 1.90 | 1.24 |
| 750 | $0.32^{+47\%}_{-30\%}$ | $0.64^{+24\%}_{-20\%}$ | 2.03 | $2.6^{+46\%}_{-30\%}$ | $4.9^{+22\%}_{-20\%}$ | $6.1^{+8.7\%}_{-11\%}$ | 1.90 | 1.24 |
| 800 | $0.20^{+47\%}_{-30\%}$ | $0.41^{+24\%}_{-20\%}$ | 2.01 | $1.90^{+46\%}_{-30\%}$ | $3.6^{+22\%}_{-20\%}$ | $4.5^{+8.7\%}_{-11\%}$ | 1.90 | 1.24 |
| 850 | $0.135^{+47\%}_{-30\%}$ | $0.27^{+24\%}_{-20\%}$ | 2.00 | $1.42^{+47\%}_{-30\%}$ | $2.7^{+22\%}_{-20\%}$ | $3.3^{+8.7\%}_{-11\%}$ | 1.89 | 1.24 |
| 900 | $0.090^{+47\%}_{-30\%}$ | $0.180^{+24\%}_{-20\%}$ | 2.00 | $1.07^{+47\%}_{-30\%}$ | $2.0^{+22\%}_{-20\%}$ | $2.5^{+8.5\%}_{-11\%}$ | 1.89 | 1.24 |
| 950 | $0.061^{+48\%}_{-30\%}$ | $0.120^{+23\%}_{-20\%}$ | 1.98 | $0.81^{+47\%}_{-30\%}$ | $1.53^{+22\%}_{-20\%}$ | $1.90^{+8.6\%}_{-11\%}$ | 1.89 | 1.24 |
| 1000 | $0.041^{+48\%}_{-30\%}$ | $0.081^{+24\%}_{-20\%}$ | 1.95 | $0.62^{+47\%}_{-30\%}$ | $1.17^{+22\%}_{-20\%}$ | $1.45^{+8.6\%}_{-11\%}$ | 1.89 | 1.24 |
| 1050 | $0.029^{+48\%}_{-30\%}$ | $0.056^{+23\%}_{-20\%}$ | 1.96 | $0.47^{+47\%}_{-30\%}$ | $0.90^{+22\%}_{-20\%}$ | $1.12^{+8.6\%}_{-11\%}$ | 1.89 | 1.24 |
| 1100 | $0.0199^{+49\%}_{-30\%}$ | $0.039^{+24\%}_{-20\%}$ | 1.94 | $0.37^{+48\%}_{-30\%}$ | $0.69^{+22\%}_{-20\%}$ | $0.86^{+8.7\%}_{-11\%}$ | 1.89 | 1.24 |
| 1150 | $0.0139^{+49\%}_{-30\%}$ | $0.027^{+24\%}_{-20\%}$ | 1.92 | $0.28^{+48\%}_{-30\%}$ | $0.54^{+22\%}_{-20\%}$ | $0.67^{+8.7\%}_{-11\%}$ | 1.90 | 1.24 |
| 1200 | $0.0098^{+49\%}_{-31\%}$ | $0.0186^{+24\%}_{-20\%}$ | 1.90 | $0.22^{+48\%}_{-30\%}$ | $0.42^{+22\%}_{-20\%}$ | $0.52^{+8.7\%}_{-12\%}$ | 1.90 | 1.24 |
| 1250 | $0.0070^{+49\%}_{-31\%}$ | $0.0130^{+25\%}_{-20\%}$ | 1.86 | $0.173^{+48\%}_{-31\%}$ | $0.33^{+22\%}_{-20\%}$ | $0.41^{+8.6\%}_{-12\%}$ | 1.90 | 1.24 |

Table 2: Best prediction $\Sigma^{\text{EFT-improved (1), NNLO}}$ for the inclusive cross sections at different $p_\perp$ cuts of phenomenological interest, and using two different prescriptions for the uncertainty (see text for details).

| $p_\perp^{\text{cut}}$ | $\text{NNLO}_{\text{quad.unc.}}^{\text{approximate}}$ [fb] | $\text{NNLO}_{\text{lin.unc.}}^{\text{approximate}}$ [fb] |
|---|---|---|
| 400 GeV | $30.7^{+9.6\%}_{-11.8\%}$ | $30.7^{+11.9\%}_{-14.2\%}$ |
| 430 GeV | $21.2^{+9.6\%}_{-11.8\%}$ | $21.2^{+11.9\%}_{-14.2\%}$ |
| 450 GeV | $16.7^{+9.5\%}_{-11.8\%}$ | $16.7^{+11.9\%}_{-14.2\%}$ |

uncertainty $\delta_{\text{NNLO}, m_t}$ as

$$\delta_{\text{NNLO}, m_t} \equiv \frac{\delta\Sigma^{\text{SM, NLO}} - \delta\Sigma^{\text{improved(0), NLO}}}{\delta\Sigma^{\text{EFT, NLO}}} \times \delta\Sigma^{\text{EFT, NNLO}}$$

$$= \frac{\delta\Sigma^{\text{SM, NLO}} - \delta\Sigma^{\text{improved(0), NLO}}}{\delta\Sigma^{\text{improved(0), NLO}}} \times \delta\Sigma^{\text{improved(0), NNLO}}. \tag{5}$$

Here, $\delta\Sigma$ refers to the perturbative correction at a given order in QCD perturbation theory, namely $\delta\Sigma^{\text{X, (N)NLO}} = \Sigma^{\text{X, (N)NLO}} - \Sigma^{\text{X, (N)LO}}$.

- The final uncertainty is obtained by combining the scale and mass effect uncertainties defined in the previous two items. In Table 2 we report the results for the cross sections, where the uncertainties are either combined in quadrature ($\text{NNLO}_{\text{quad.unc.}}^{\text{approximate}}$) or summed linearly ($\text{NNLO}_{\text{lin.unc.}}^{\text{approximate}}$). In the following, we work under the assumption that the three sources of uncertainty are uncorrelated, and therefore will consider the combination in quadrature as our central prescription.

- An additional source of uncertainty is given by the top-mass scheme, for which we adopt the on-shell scheme used in the calculation of refs. [15,17]. The difference between the on-shell and the $\overline{\text{MS}}$ scheme has been recently studied in ref. [16],[2] and shown to be substantial at LO for typical renormalisation scales in boosted Higgs production. In the same reference it was shown that differences are reduced at NLO, although they are still substantial and warrant careful consideration. We leave this discussion for future work of the working group.

In Fig. 2 we show the cumulative cross section as a function of the $p_\perp$ cut. The figure compares the NNLO EFT, Born-improved NNLO EFT (EFT-improved(0)) and our best prediction (EFT-improved(1)), obtained using Eq. (4). Fig. 3 shows the ratio of the latter two predictions to the central value of the EFT-improved(1) prediction. The uncertainties in the EFT-improved(0) band has been obtained by pure scale variation, while the uncertainty in the EFT-improved(1) prediction is estimated as outlined above.

## 2.2 Event generators

In this section we report the predictions obtained with different event generators for the boosted-Higgs scenario.

We compare the following Monte-Carlo tools:

- `POWHEG gg_h` [18]: NLO accurate for inclusive gluon fusion and LO ($\mathcal{O}(\alpha_s^3)$) in the $p_\perp$ spectrum. The calculation is performed in the heavy-top EFT. The default POWHEG $\mu_R$ and $\mu_F$ scales are used. The `hfact` parameter [18] is set to $h = 104$ GeV as in the CMS analysis [1] (this only impacts the predictions matched to a parton shower below).

---

[2]The top mass scheme was also discussed in Ref. [30].

- POWHEG HJ [19]: NLO accurate ($\mathcal{O}(\alpha_s^4)$) in the Higgs $p_\perp$ spectrum. The calculation is performed in the heavy-top EFT. $\mu_R$ and $\mu_F$ are set to $H_T/2 = 1/2\left(\sqrt{m_H^2 + p_\perp^2} + \sum_{i=1}^n |p_{t,i}|\right)$, where $p_{t,i}$ is the transverse momentum of the $i$-th radiated parton (n = 1 for Born/Virtual events, n = 2 for real events).

- HJ-MiNLO [20]: NLO for inclusive gluon fusion and NLO in the $p_\perp$ spectrum. $\mu_R$ and $\mu_F$ are always set to $p_\perp$. The calculation is performed in the heavy-top EFT, but finite $m_t$ effects can be included via a rescaling by the LO spectrum in the full SM. Born events with one jet terms are proportional to $\alpha_s^2(m_H)\alpha_s(p_\perp)$, while NLO corrections are proportional to $\alpha_s^2(m_H)\alpha_s^2(p_\perp)$.

- MG5_MC@NLO [21,31]: predictions obtained by merging samples of 0,1, and 2 jets, NLO accurate for all the above multiplicities. Finite $m_t$ corrections are included exactly in the Born and real corrections for all multiplicities, and approximately in the virtual corrections by rescaling the heavy-top EFT virtual corrections by the LO result in the full SM. The scale is set following the FxFx [32] prescription and the merging scale is set to 30 GeV. The merging scale sets the effective momentum scale at which the event sample transitions between the various jet-multiplicities, cf. Section 2 of Ref. [32].

The results for the POWHEG/MiNLO generators are reported both at fixed order and matched to the Pythia 6 parton shower Monte Carlo [33], in Table 3 and 4, respectively. Table 3 shows the predictions from the POWHEG/MiNLO generators before the matching to a parton shower is performed, while Table 4 reports the predictions matched to a parton shower simulation. The last row of the tables shows the result of HJ-MiNLO including mass effects, as implemented in ref. [27]. The results include only the top contribution, implemented through a rescaling of the EFT result by the exact LO spectrum, and hence very similar in spirit to the prescription introduced in Section 2.1, in Eq. (3). In the large Higgs transverse momentum region, the generator HJ-MiNLO reproduces exactly the NNLOPS [22] and MiNNLO$_{PS}$ [34,35] generators currently used in Higgs analyses for the gluon fusion channel at the LHC. Uncertainties are obtained through a 7-point scale variation around the central renormalisation and factorisation scales by a factor of two.

By inspecting the last two rows of Tables 3 and 4, we observe that the inclusion of the

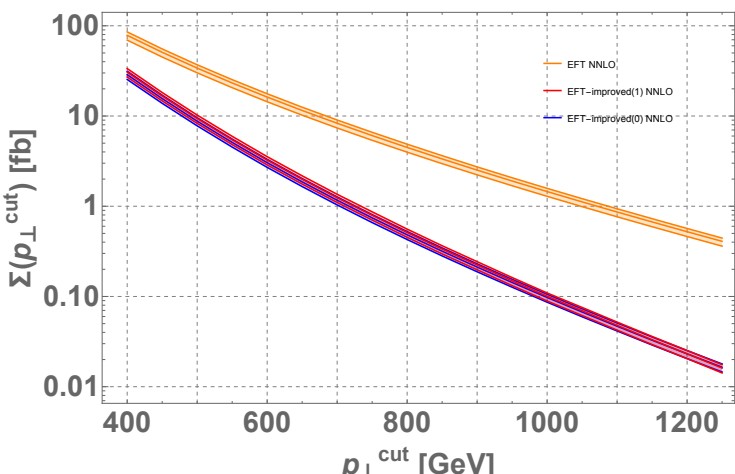

Figure 2: Cumulative cross section as a function of the $p_\perp$ cut at NNLO in the heavy-top EFT, as well as rescaled by the LO (NLO) full-SM spectrum labelled by EFT-improved(0) (EFT-improved(1)). See the text for description. The ratio of the EFT-improved(1) and EFT-improved(0) predictions is shown in Fig. 3.

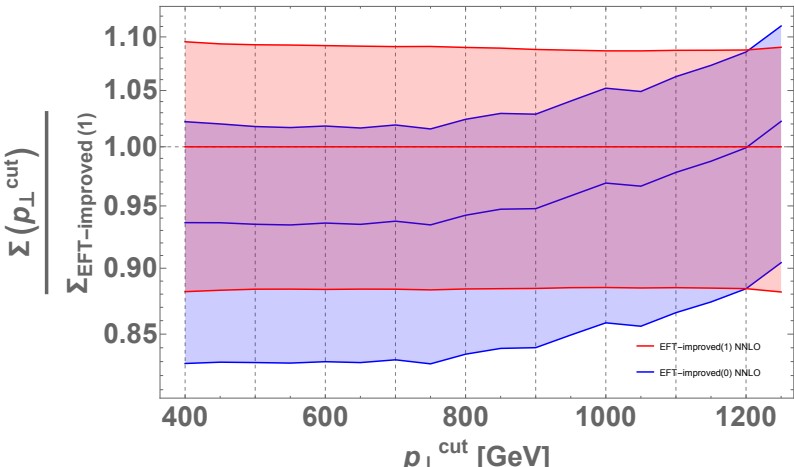

Figure 3: Ratio of the cumulative cross section as defined in the EFT-improved(0) and EFT-improved(1) approximation (see the text for description) to the central value of the EFT-improved(1) result as a function of the $p_\perp$ cut.

Table 3: Results from the indicated event generators for different $p_\perp$ cuts (in GeV units) before the matching to parton showers is performed (labelled as *Fixed order level* in the table). Predictions are expressed in [pb] units. The total cross section for $gg \rightarrow H$ obtained with the indicated event generator is also reported whenever available. The total cross section shown here is by construction identical to the one reported in Table 4 - including the scale uncertainties.

| Fixed order level [pb] | Total | $p_\perp^{\text{cut}} > 400$ GeV | $p_\perp^{\text{cut}} > 450$ GeV | $p_\perp^{\text{cut}} > 500$ GeV |
|---|---|---|---|---|
| $\text{ggh}^{\text{hfact}=104}_{m_t=\infty}$ | 30.3 | 0.0730 | 0.0507 | 0.0362 |
| HJ $m_t = \infty$, 5 GeV gen. cut | – | 0.0643 | 0.0413 | 0.0278 |
| HJ $m_t = \infty$, 50 GeV gen. cut | – | 0.0644 | 0.0416 | 0.0277 |
| HJ-MiNLO $m_t = \infty$ | 32.1 | 0.0778 | 0.0509 | 0.0343 |
| HJ-MiNLO $m_t = 171.3$ GeV | 33.8 | 0.0281 | 0.0153 | 0.0089 |

parton shower has a moderate impact on the result (at the $2 - 5\%$ level), as one expects for the considered kinematics regime.

The results obtained with MG5_MC@NLO are obtained with top mass corrections included exactly in the Born and real corrections, and approximately in the virtual corrections by rescaling the EFT virtual corrections by the LO result in the full SM. Exact bottom quark mass effects are not included as they are negligible in the considered region. The events are showered with the Pythia 8 parton shower Monte Carlo [36]. The results for some relevant $p_\perp$ cuts are summarised in Table 5, together with a comparison to the results of the HJ-MiNLO generator, and to our best prediction described in Section 2.1. The quoted uncertainties have been obtained by a 9-point scale variation, i.e. independently around the central renormalisation and factorisation scales by a factor of two.[3]

We observe that the predictions obtained with the more accurate generators used in the study (HJ-MiNLO and MG5_MC@NLO) are in very good agreement with one another. Moreover, they both reproduce, within uncertainties, the best prediction obtained in the previous section.

---

[3]The uncertainty prescriptions adopted in Table 5 reflect the nominal prescriptions used in Refs. [20, 21]. For the fixed order prediction we adopt the prescription discussed in the Section 2.1.

Table 4: Results matched to parton shower for different $p_\perp$ cuts (in GeV units) for the indicated event generators. Predictions are expressed in [pb] units. The total cross section for $gg \to H$ obtained with the indicated event generator is also reported whenever available.

| Parton shower matched level [pb] | Total | $p_\perp^{\text{cut}} > 400$ GeV | $p_\perp^{\text{cut}} > 450$ GeV | $p_\perp^{\text{cut}} > 500$ GeV |
|---|---|---|---|---|
| $\mathrm{ggh}_{m_t=\infty}^{\text{hfact}=104}$ | $30.3^{+6.1}_{-4.7}$ | $0.0829^{+0.0451}_{-0.0266}$ | $0.0577^{+0.0325}_{-0.019}$ | $0.0408^{+0.0236}_{-0.0137}$ |
| HJ $m_t = \infty$, 5 GeV gen. cut | $-$ | $0.0651^{+0.0156}_{-0.0131}$ | $0.0417^{+0.01}_{-0.0084}$ | $0.0279^{+0.0067}_{-0.0057}$ |
| HJ $m_t = \infty$, 50 GeV gen. cut | $-$ | $0.0651^{+0.0156}_{-0.0131}$ | $0.0418^{+0.01}_{-0.0085}$ | $0.0278^{+0.0066}_{-0.0056}$ |
| HJ-MiNLO $m_t = \infty$ | $32.1^{+11}_{-4.9}$ | $0.0803^{+0.9087}_{-0.0164}$ | $0.0524^{+0.0118}_{-0.0107}$ | $0.0353^{+0.0078}_{-0.0072}$ |
| HJ-MiNLO $m_t = 171.3$ GeV | $33.8^{+11.4}_{-5.2}$ | $0.029^{+0.007}_{-0.006}$ | $0.0161^{+0.0036}_{-0.0033}$ | $0.0091^{+0.0021}_{-0.0018}$ |

Table 5: Comparison of predictions at fixed order in the $\Sigma^{\text{EFT-improved (1), NNLO}}$ approximation, with HJ-MINLO and with MG5_MC@NLO. The uncertainties in the three predictions are obtained by means of a 7 points scale variation ($\text{NNLO}^{\text{approximate}}_{\text{quad.unc.}}$ and HJ-MINLO), and 9 point scale variation (MG5_MC@NLO), respectively. The difference in the uncertainty prescription is reflected in the different theoretical errors quoted in the table. See text for more details.

| $p_\perp^{\text{cut}}$ | $\text{NNLO}^{\text{approximate}}_{\text{quad.unc.}}$ [fb] | HJ-MINLO [fb] | MG5_MC@NLO [fb] |
|---|---|---|---|
| 400 GeV | $30.7^{+9.6\%}_{-11.8\%}$ | $29^{+24\%}_{-21\%}$ | $31.5^{+31\%}_{-25\%}$ |
| 450 GeV | $16.7^{+9.5\%}_{-11.8\%}$ | $16.1^{+22\%}_{-21\%}$ | $17.1^{+31\%}_{-25\%}$ |

We conclude that the above two generators can be safely used to perform accurate studies in the boosted regime. However, state of the art QCD predictions reach a higher level of precision and novel methods are necessary to exploit such calculations in the context of Monte Carlo simulations.

## 3 Predictions for other production modes

In this Section we report the breakdown of the boosted Higgs cross section into different production channels. In the following we consider both QCD and EW perturbative corrections. We start by discussing the former, for which we consider the same YR4 setup [23] discussed in Section 2.1 unless stated otherwise. For vector boson fusion (VBF), the prediction is obtained from refs. [37], where the VBF cross section is computed to NNLO accuracy in perturbative QCD ($\mathcal{O}(\alpha_s^2)$) obtained in the so called *factorised* approximation [38]. In the same approximation, N³LO corrections are known [39], but are negligible for the accuracy considered in this work. Non-factorising corrections have been recently estimated [40,41], and it was concluded that they may be potentially relevant in the considered phase space region. Nevertheless, we do not expect these corrections to affect our qualitative conclusions. For this process we set the renormalisation and factorisation scales to $\mu_R^2 = \mu_F^2 = m_H/2\sqrt{(m_H/2)^2 + p_\perp^2}$. Perturbative uncertainties are obtained by varying both scales by a factor of two while keeping $\mu_R = \mu_F$ (3-point variation). For associated production VH ($V = W^\pm, Z$), we consider NLO ($\mathcal{O}(\alpha_s)$)

Table 6: Predictions for the cumulative Higgs boson cross section as a function of the lower $p_\perp$ cut (the quoted gluon fusion cross section is obtained in the $\Sigma^{\text{EFT-improved (1), NNLO}}$ approximation). We show QCD predictions for the various channels contributing to Higgs production. The table does not contain the EW corrections.

| $p_\perp^{\text{cut}}$[GeV] | $\Sigma_{\text{ggF}}^{\text{NNLO}^{\text{approximate}}_{\text{quad.unc.}}}(p_\perp^{\text{cut}})$ [fb] | $\Sigma_{\text{VBF}}^{\text{NNLO}}(p_\perp^{\text{cut}})$ [fb] | $\Sigma_{\text{VH}}^{\text{NLO}}(p_\perp^{\text{cut}})$ [fb] | $\Sigma_{\text{t\bar{t}H}}^{\text{NLO}}(p_\perp^{\text{cut}})$ [fb] |
|---|---|---|---|---|
| 400 | $30.67^{+9.59\%}_{-11.84\%}$ | $14.23^{+0.15\%}_{-0.19\%}$ | $11.16^{+4.12\%}_{-3.68\%}$ | $6.89^{+12.62\%}_{-12.97\%}$ |
| 450 | $16.70^{+9.53\%}_{-11.76\%}$ | $8.06^{+0.24\%}_{-0.23\%}$ | $6.87^{+4.6\%}_{-3.49\%}$ | $4.24^{+12.84\%}_{-13.15\%}$ |
| 500 | $9.41^{+9.44\%}_{-11.72\%}$ | $4.75^{+0.33\%}_{-0.29\%}$ | $4.39^{+4.43\%}_{-4.04\%}$ | $2.66^{+12.85\%}_{-13.22\%}$ |
| 550 | $5.46^{+9.43\%}_{-11.69\%}$ | $2.90^{+0.34\%}_{-0.36\%}$ | $2.87^{+4.44\%}_{-3.74\%}$ | $1.76^{+14.23\%}_{-13.93\%}$ |
| 600 | $3.25^{+9.31\%}_{-11.64\%}$ | $1.82^{+0.41\%}_{-0.39\%}$ | $1.91^{+5.22\%}_{-4.71\%}$ | $1.11^{+12.99\%}_{-13.4\%}$ |
| 650 | $1.99^{+9.21\%}_{-11.63\%}$ | $1.17^{+0.49\%}_{-0.39\%}$ | $1.30^{+4.67\%}_{-4.28\%}$ | $0.72^{+12.6\%}_{-13.26\%}$ |
| 700 | $1.24^{+9.09\%}_{-11.57\%}$ | $0.77^{+0.57\%}_{-0.45\%}$ | $0.90^{+4.15\%}_{-5.4\%}$ | $0.47^{+11.42\%}_{-12.74\%}$ |
| 750 | $0.79^{+9.16\%}_{-11.60\%}$ | $0.51^{+0.69\%}_{-0.56\%}$ | $0.62^{+5.15\%}_{-4.66\%}$ | $0.32^{+11.53\%}_{-12.84\%}$ |
| 800 | $0.51^{+9.05\%}_{-11.56\%}$ | $0.35^{+0.71\%}_{-0.6\%}$ | $0.44^{+5.64\%}_{-4.13\%}$ | $0.22^{+11.42\%}_{-13.3\%}$ |

predictions obtained with the POWHEG-BOX-V2 [42, 43].[4] The scales are set to the invariant mass of the $VH$ system as $\mu_R = \mu_F = \sqrt{(p_H + p_V)^2}$, and perturbative uncertainties are again obtained by varying both scales by a factor of two while keeping $\mu_R = \mu_F$ (3-point variation). Also in this case NNLO corrections are known to be small, with the exception of the contribution from gluon fusion [23]. Therefore, we do not include them in the following. For VBF and VH we use a 3- rather than 7-point variation, because the latter has been found to be almost entirely contained within the former [37, 44]. We have explicitly verified that this is the case in the boosted regime considered here. Finally, for t$\bar{t}$H, we consider NLO ($\mathcal{O}(\alpha_s^3)$) predictions obtained with Sherpa+OpenLoops [45, 46]. In this case the perturbative scales are set to $\mu_R = \mu_F = (M_{T,t} + M_{T,\bar{t}} + M_{T,H})/2$, and uncertainties are obtained with a 7-point variation.

The results are reported in Table 6. We stress that the quoted uncertainty only accounts for QCD scale variations estimated as outlined above, and it does not contain PDF and $\alpha_s$ errors.

For all channels but gluon fusion, NLO EW corrections have been known for some time (cf. ref. [23, 47–54]), and are obtained here using Sherpa+OpenLoops [45, 46, 55–57]. The emission of weak gauge bosons is not included in the EW corrections, and should be considered as separate background reactions. We report the results in Table 7, which displays the percentage decrease of the corresponding cross sections of Table 6 due to the inclusion of electro-weak corrections. The calculation of the EW corrections in the gluon fusion channel has recently been considered in refs. [58–60]. Although a complete calculation in the regime considered in this work is not yet available, we stress that these corrections are expected to be sizeable at large transverse momentum, and must be estimated for an accurate prediction of the gluon-fusion production rate. Moreover, we observe that this observable receives substantial contributions from other production modes, which therefore must be taken into account together with the gluon-fusion channel in experimental analyses.

Finally, the absolute and relative contributions of the different production modes up to transverse momenta of 1.25 TeV are summarised in Fig. 4, including both QCD and EW corrections.

---

[4]We note that, in addition, the $ZH$ channel may receive large perturbative correction to the gluon-induced subprocess.

Table 7: Percentage decrease of the cross sections of Table 6 due to the inclusion of electro-weak corrections as a function of the cut in $p_\perp$.

| $p_\perp^{\text{cut}}[\text{GeV}]$ | VBF | VH | $t\bar{t}H$ |
|---|---|---|---|
| 400 | −17.80% | −19.05% | −6.95% |
| 450 | −19.43% | −20.83% | −7.75% |
| 500 | −21.05% | −22.50% | −8.49% |
| 550 | −22.34% | −24.07% | −9.11% |
| 600 | −23.73% | −25.56% | −9.91% |
| 650 | −25.03% | −26.98% | −10.67% |
| 700 | −26.29% | −28.30% | −11.37% |
| 750 | −27.35% | −29.60% | −11.94% |
| 800 | −28.42% | −30.83% | −12.51% |

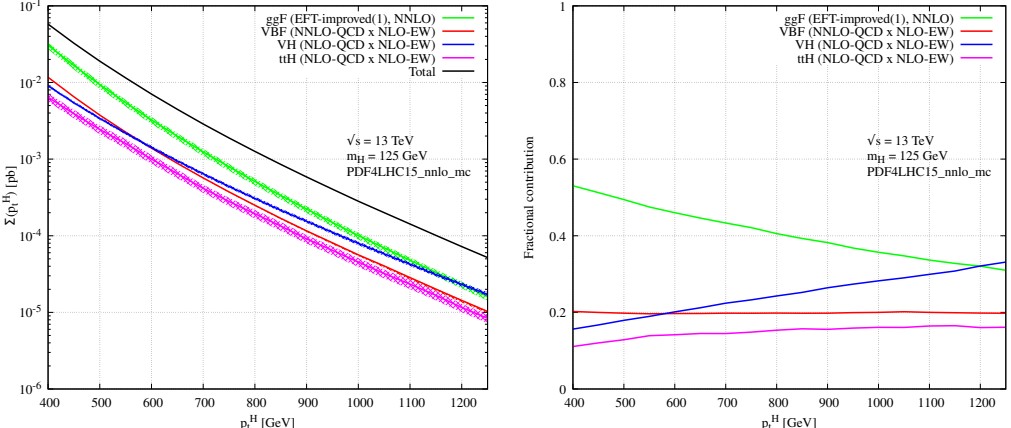

Figure 4: Cumulative cross section for the production of a Higgs boson as a function of the Higgs boson transverse momentum cut. The cross section due to the gluon-fusion (green), VBF (red), vector boson associated (blue) and top-quark pair associated (magenta) production mode are shown in absolute values (left) and relative size (right).

## 4 Summary and conclusions

In this article we studied the inclusive production of a boosted Higgs boson at the LHC. We presented a combination of accurate QCD predictions for the various production channels, and provided a recommendation for the cumulative distribution at large transverse momenta in the gluon-fusion channel. The resulting predictions are reported in Table 6 for different values of the lower cut on the Higgs transverse momentum. The table shows that in the boosted regime the dominance of the gluon-fusion channel is much less significant, and a consistent inclusion of different production modes is necessary. This is even more important in view of BSM interpretations since different channels can be affected differently by new-physics effects. It is therefore desirable in experimental analyses to avoid subtracting different Higgs production channels from the experimental measurement as a way of assessing the gluon-fusion contribution. Such a subtraction can only be done under strong theoretical assumptions. An unbiased way of reporting the experimental results necessarily involves quoting the fiducial cross sections.

For the gluon fusion contribution, we compare the resulting predictions to those of Monte-Carlo event generators in Table 5 and find good agreement within the quoted uncertainties. This implies that one can safely use the predictions from the considered event generators with the associated theoretical errors in the simulation of the boosted Higgs cross section. Additional values of the gluon-fusion cross section are also reported in Appendix A up to scales of 1.25 TeV.

We stress that we did not account here for other sources of theoretical uncertainties (such as the top mass scheme, PDF and couplings uncertainties, and EW corrections to the gluon-fusion process), which must be included in the overall systematics in phenomenological studies of the boosted Higgs cross section.

# Acknowledgments

This work was done within the LHC Higgs Working Group (LHCHWG).

**Funding information** K.H. was supported by the Science and Technology Facilities Council (STFC) under grant award ST/P000274/1, and by the European Commission through the ERC Consolidator Grant HICCUP (No. 614577).

Table 8: Gluon fusion predictions for the cumulative Higgs boson cross section obtained in the $\Sigma^{\text{EFT-improved (1), NNLO}}$ approximation as a function of the lowest allowed $p_\perp$.

| $p_\perp^{\text{cut}}$[GeV] | $\Sigma_{\text{ggF}}^{\text{NNLO}^{\text{approximate}}_{\text{quad.unc.}}}(p_\perp^{\text{cut}})$ [fb] |
|---|---|
| 400 | $30.67^{+9.59\%}_{-11.84\%}$ |
| 410 | $27.03^{+9.59\%}_{-11.80\%}$ |
| 420 | $23.94^{+9.54\%}_{-11.77\%}$ |
| 430 | $21.23^{+9.55\%}_{-11.77\%}$ |
| 440 | $18.83^{+9.54\%}_{-11.78\%}$ |
| 450 | $16.70^{+9.53\%}_{-11.76\%}$ |
| 460 | $14.79^{+9.45\%}_{-11.73\%}$ |
| 470 | $13.20^{+9.46\%}_{-11.73\%}$ |
| 480 | $11.81^{+9.51\%}_{-11.75\%}$ |
| 490 | $10.53^{+9.51\%}_{-11.73\%}$ |
| 500 | $9.41^{+9.44\%}_{-11.72\%}$ |
| 510 | $8.44^{+9.48\%}_{-11.72\%}$ |
| 520 | $7.56^{+9.47\%}_{-11.71\%}$ |
| 530 | $6.76^{+9.41\%}_{-11.68\%}$ |
| 540 | $6.06^{+9.39\%}_{-11.66\%}$ |
| 550 | $5.46^{+9.43\%}_{-11.69\%}$ |
| 560 | $4.92^{+9.44\%}_{-11.71\%}$ |
| 570 | $4.43^{+9.37\%}_{-11.70\%}$ |
| 580 | $3.99^{+9.36\%}_{-11.70\%}$ |
| 590 | $3.59^{+9.34\%}_{-11.65\%}$ |

# A  Gluon fusion cross section up to 1.25 TeV

In this appendix we report additional predictions for the gluon fusion channel. Table 8 shows results in the range $p_\perp^{\text{cut}} \in [400, 600]$ GeV for a finer binning than the one considered in the text, while cross section up to $p_\perp^{\text{cut}} = 1.25$ TeV in 50-GeV bins are summarised in Table 9.

Table 9: Gluon fusion cross section obtained in the $\Sigma^{\text{EFT-improved (1), NNLO}}$ approximation in highly boosted regime.

| $p_\perp^{\text{cut}}$[GeV] | $\Sigma_{\text{ggF}}^{\text{NNLO}_{\text{quad.unc.}}^{\text{approximate}}}(p_\perp^{\text{cut}})$ [fb] |
|---|---|
| 400 | $30.67^{+9.59\%}_{-11.84\%}$ |
| 450 | $16.70^{+9.53\%}_{-11.76\%}$ |
| 500 | $9.41^{+9.44\%}_{-11.72\%}$ |
| 550 | $5.46^{+9.43\%}_{-11.69\%}$ |
| 600 | $3.25^{+9.31\%}_{-11.64\%}$ |
| 650 | $1.99^{+9.21\%}_{-11.63\%}$ |
| 700 | $1.24^{+9.09\%}_{-11.57\%}$ |
| 750 | $0.80^{+9.16\%}_{-11.60\%}$ |
| 800 | $0.51^{+9.05\%}_{-11.56\%}$ |
| 850 | $0.34^{+8.93\%}_{-11.58\%}$ |
| 900 | $0.22^{+8.81\%}_{-11.56\%}$ |
| 950 | $0.15^{+8.74\%}_{-11.50\%}$ |
| 1000 | $0.10^{+8.68\%}_{-11.49\%}$ |
| 1050 | $0.07^{+8.68\%}_{-11.53\%}$ |
| 1100 | $0.05^{+8.73\%}_{-11.50\%}$ |
| 1150 | $0.03^{+8.73\%}_{-11.54\%}$ |
| 1200 | $0.02^{+8.76\%}_{-11.58\%}$ |
| 1250 | $0.02^{+9.02\%}_{-11.84\%}$ |

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
