# Peer review of "Precise predictions for boosted Higgs production"

_SciPost Physics Core, doi:SciPost Phys. Core 7, 001 (2024)_

## Round 1 · Referee Report · Anonymous (Referee 1) · 2023-8-11

Strengths

  1. The paper focuses on an important channel for studying the Higgs boson
  2. The paper provides clear and well supported recommendations for appropriately handling the theoretical predictions by experimentalists
  3. The paper has all the details needed to reproduce the results and very clearly lays out all assumptions and approximations made

Weaknesses

  1. There are some inconsistencies in how scale variations are handled between different predictions with no reasoning to support this.

Report

Overall, the paper covers a very important and relevant topic for the LHC. The paper provides a detailed study of the precision for the inclusive rate of Higgs boson production in a boosted regime. They investigate how the rates change as a function of the minimum transverse momentum cut. I would recommend this paper for publication with a few additional modifications listed below.

Requested changes

  1. While I agree that it is "well known that the pure EFT computation fails to describe the $p_\perp$ spectrum for transverse momenta larger than ~200 GeV," I believe the authors should still supply a reference for less familiar readers.
  2. In Figure 1, the authors have uncertainty bands associated with scale variations and simply mention that it is the 7-point scale. It is not until later in the paper that this scheme is introduced. I recommend that this point be moved earlier in the paper to before the first reference to this scale.
  3. In the comparison to event generators for gg fusion, Sherpa is not included. Is there an explanation for why Sherpa is not included in this section but is included in the later sections for non-gluon fusion processes and EW corrections? Since Sherpa is another major event generator, I think it is important to include it in the gluon fusion comparisons.
  4. In Table 3, only the total column has uncertainties. This is the only table that does not include uncertainties on all the numbers. The authors should add those values into the table.
  5. When discussing the results from MG5_MC@NLO, the authors switch to a 9-point scale with no motivation for the change just for MadGraph. This should either be justified or changed to the 7-point scale for consistency.
  6. Additionally, when discussing the VH production the authors only consider the 3-point scale variation without any justification for not using the 7-point scale. Again they should either justify this choice or use the 7-point scale.
  7. In the caption of Figure 4, it is unclear what is mean by "as a function of the lowest Higgs boson transverse momentum." I think it would be made more clear to change to "as a function of the Higgs boson transverse momentum cut".

---

## Round 1 · Referee Report · Anonymous (Referee 2) · 2023-8-22

Strengths

1- The paper presents a detailed study of the Higgs boson production in the important kinematic regime of large transverse momenta.

2- The paper provides a detailed comparison with the state-of-the-art theoretical predictions. The comparison can be very useful for experimental analyses.

3- The paper provides an estimate of theoretical uncertainties from missing higher orders and from finite top mass effects.

4- The paper explains the advantages and limitations of the various theoretical predictions.

Weaknesses

1- The originality of the paper is not particularly high. A deeper and broader study of the comparison between theoretical predictions would have been useful for increasing the relevance of the manuscript.

2- In some points the authors introduce different prescriptions for scale choices (central values and variation) without justificationsor comments.

3- There are some sentences that may be obscure to a non expert reader.

Report

The paper covers an interesting an important topic for the physics of the Large Hadron Collider. Higher-order corrections are essential to provide
accurate theoretical predictions to be compared with precise
experimental data.

The results are interesting and very useful. The paper is well written and very well organised. In my opinion it deserves to be published with minor corrections. I give below some suggestions for improving the manuscript.

Requested changes

1- I suggest to motivate the use of a given scale choice. It could be sufficient to provide references where the choice has been motivated.

2- The authors should clearly describe the technical aspect (e.g. the meaning of 3, 7 and 9 point scale variation or the merging scale). Again references to the literature should suffice.

3- I suggest complete Table 3 with the missing uncertainties and Table 5 with the missing entry.

---

## Round 2 · Referee Report · Anonymous (Referee 1) · 2023-11-22

Report

Overall, the paper covers a very important and relevant topic for the LHC. The authors have adequately addressed my previous concerns in their updated draft. While I would have liked to see Sherpa results included along side the other event generators to have all the relevant LHC tools, I understand the constraints on the authors. I look forward to a more detailed study including these missing pieces. With these changes, I recommend the paper for publication.

---

## Round 2 · Referee Report · Anonymous (Referee 2) · 2023-11-22

Strengths

1- The paper presents a detailed study of the Higgs boson production in the important kinematic regime of large transverse momenta.

2- The paper provides a detailed comparison with the state-of-the-art theoretical predictions. The comparison can be very useful for experimental analyses.

3- The paper provides an estimate of theoretical uncertainties from missing higher orders and from finite top mass effects.

4- The paper explains the advantages and limitations of the various theoretical predictions.

Weaknesses

1- The originality of the paper is not particularly high. A deeper and broader study of the comparison between theoretical predictions would have been useful for increasing the relevance of the manuscript.

Report

The authors have modified the manuscript accordingly with the requirements of the previous report.

Requested changes

I do not have additional changes to request.

---

## Round 2 · Author Response

Referee 1

1- I suggest to motivate the use of a given scale choice. It could be sufficient to provide references where the choice has been motivated.

2- The authors should clearly describe the technical aspect (e.g. the meaning of 3, 7 and 9 point scale variation or the merging scale). Again references to the literature should suffice.

We agree with both points 1 and 2, and have added a number of references and text throughout the paper to address these comments.

3- I suggest complete Table 3 with the missing uncertainties and Table 5 with the missing entry.

The fixed order predictions do no contain uncertainties as this is rather costly requiring 7 independent runs. The total of table 3, on the other hand, is identical to the total in table 4, hence the inclusion of uncertainties on the total. To make table 3 more consistent, we have removed the uncertainties on the total, and have added a sentence to the captions in table 3 and 4 stressing that it is only table 4 which contains the recommended predictions.

The 430 GeV point in table 5 was added at a late stage in the write-up of the manuscript. The HJ-MINLO authors did not provide this value due to time constraints. For simplicity we have removed that 430 GeV row.

Referee 2

  1. While I agree that it is "well known that the pure EFT computation fails to describe the pt spectrum for transverse momenta larger than ~200 GeV," I believe the authors should still supply a reference for less familiar readers.

We have added a reference to this effect.

  1. In Figure 1, the authors have uncertainty bands associated with scale variations and simply mention that it is the 7-point scale. It is not until later in the paper that this scheme is introduced. I recommend that this point be moved earlier in the paper to before the first reference to this scale.

We have added the definition to the caption.

  1. In the comparison to event generators for gg fusion, Sherpa is not included. Is there an explanation for why Sherpa is not included in this section but is included in the later sections for non-gluon fusion processes and EW corrections? Since Sherpa is another major event generator, I think it is important to include it in the gluon fusion comparisons.

For the EW corrections and ttH production Sherpa has only been used as a fixed order generator. At this level the gluon fusion prediction agrees with the standard NLO result. The Sherpa authors were originally contacted when we started this project, but were unfortunately not able to provide any predictions before the note was finished. We agree with the referee that providing Sherpa predictions would be ideal. For this reason we are currently planning on setting up a more extensive study across all the sub-groups in the LHCHWG, taking into account recent progress in for instance EW corrections to ggF as well as a robust assessment of the top-mass scheme uncertainty. We will include predictions from Sherpa along with other major event generators in this study.

  1. In Table 3, only the total column has uncertainties. This is the only table that does not include uncertainties on all the numbers. The authors should add those values into the table.

The fixed order predictions do no contain uncertainties as this is rather costly requiring 7 independent runs. The total of table 3, on the other hand, is identical to the total in table 4, hence the inclusion of uncertainties on the total. To make table 3 more consistent, we have removed the uncertainties on the total, and have added a sentence to the captions in table 3 and 4 stressing that it is only table 4 which contains the recommended predictions.

  1. When discussing the results from MG5_MC@NLO, the authors switch to a 9-point scale with no motivation for the change just for MadGraph. This should either be justified or changed to the 7-point scale for consistency.

The apparent inconsistency in table 5 is due to the fact that we followed the uncertainty prescriptions that were used in the MiNLO and MC@NLO publications, and were recommended by the authors. We have added a footnote to clarify this.

  1. Additionally, when discussing the VH production the authors only consider the 3-point scale variation without any justification for not using the 7-point scale. Again they should either justify this choice or use the 7-point scale.

For VBF and VH the 3-point scale variation has been found in previous studies to almost entirely contain the 7-point. We have added two sentences and references to this effect.

  1. In the caption of Figure 4, it is unclear what is mean by "as a function of the lowest Higgs boson transverse momentum." I think it would be made more clear to change to "as a function of the Higgs boson transverse momentum cut".

We agree and have changed this.

---

## Round 2 · List of Changes

The changes are as described in the "Author comments". Most are minor, e.g. adding of references and clarifications in the text.

---

## Editorial Decision

published